

# Alcohol consumption and employment: a cross-sectional study of office workers and unemployed people

Simone De Sio[1], Roberta Tittarelli[2], Giuseppe Di Martino[3], Giuseppe Buomprisco[4], Roberto Perri[4], Guglielmo Bruno[4], Flaminia Pantano[5], Giulio Mannocchi[6], Enrico Marinelli[5] and Fabrizio Cedrone[3]

[1] School of Occupational Medicine—U.R. Occupational Medicine, "Sapienza" University of Rome, Rome, Italy
[2] Unit of Forensic Toxicology, Department of Anatomical, Histological, Forensic and Orthopedic Sciences, "Sapienza" University of Rome, Rome, Italy
[3] School of Hygiene and Preventive Medicine, University "G.d'Annunzio" of Chieti-Pescara, Chieti, Italy
[4] School of Occupational Medicine—U.R. Occupational Medicine, "Sapienza" University of Rome, Rome, Italy
[5] Unit of Forensic Toxicology—Department of Anatomical, Histological, Forensic and Orthopedic Sciences, "Sapienza" University of Rome, Rome, Italy
[6] Bioethics and Legal Medicine Centre, School of Law, University of Camerino, Camerino, Italy

Corresponding author
Simone De Sio,
simone.desio@uniroma1.it

## ABSTRACT

**Background**. Alcohol is a psychoactive substance with toxic and addictive properties. Biomarkers like GGT, AST, ALT and MCV are influenced by excessive ethanol consumption. Alcohol consumption represents a health risk and it has been linked to unemployment. The aim of this study how working status predict alcohol consumption through a cross sectional study comparing alcohol-related biomarkers levels in office workers and unemployed people.

**Methods**. This study includes 157 office workers and 157 unemployed people, who were recruited from January to December 2018. A propensity score matching procedure was applied to obtain two homogenous groups in terms of age and gender. A non-parametric analysis was performed on serum biomarkers that are generally altered by alcohol consumption. Logistic regression models were designed to evaluate how working status predict abnormal biomarker levels related with alcohol consumption.

**Results**. No differences in median biomarker values were found between groups. Logistic regression analysis showed that office work is a negative predictor of pathological biomarker levels. Office workers had a significant relation with the levels of GGT (OR 0.48; 95% CI [0.28–0.84]), AST (OR 0.42; 95% CI [0.22–0.78]), ALT (OR 0.39; 95% CI [0.23–0.66]), and MCV (OR 0.37; 95% CI [0.19–0.70]).

**Conclusion**. Office workers had lower absolute frequencies of pathological values of alcohol consumption biomarkers, after matching for age and gender compared with unemployed people. In addition, a significant negative association between office work is a negative predictor of biomarker levels of alcohol consumption. These results showed that work is an important determinant of health and that can represent a benefit for workers in terms of reducing the risk of consuming alcohol.

## INTRODUCTION

Alcohol is a psychoactive substance with toxic and addictive properties. Its consumption increases the risk for infectious diseases (*Taylor et al., 2016*; *Rehm et al., 2017*), non-communicable diseases (*Gao & Bataller, 2011*; *Roerecke & Rehm, 2010*; *O'Keefe et al., 2014*; *Klatsky, 2015*), and injuries (*Seedat et al., 2009*; *Mitra et al., 2017*). Alcohol has direct toxic effects on all organs of the body, including the brain. It is a psychoactive substance which causes addiction and its effects last for hours after consumption (*Babor et al., 2010*).

The harmful use of alcohol causes 3 million deaths a year, is responsible for 5.1% of the global disease burden and continues to be one of the main risk factors for illness at a global level. Despite a reduction of drinkers worldwide of about 5% from 47.6% to 43.0% since 2000, alcohol is still consumed by more than half of the population in three WHO regions, which include the European Region (*World Health Organization, 2018*).

Several studies have deepened the knowledge of the relationships between alcohol and work. They highlighted how alcohol reduces employment and increases unemployment, absenteeism and the risk for injuries. Besides, it can also negatively influence productivity and work performance (*Mullahy & Sindelar, 1996*; *Terza, 2002*; *MacDonald & Shields, 2004*; *Johansson et al., 2007*; *Ames & Bennet, 2011*). This can result in job loss, especially in a competitive job market.

Other studies have shown that the financial challenges associated with unemployment increase tension, anxiety and family discord, and this can lead to an increase in alcohol consumption (*Karasek & Töres, 1990*; *Peirce et al., 1994*; *Catalano et al., 2011*). In both working and unemployed people, alcohol consumption is often seen as a coping strategy (*Merrill & Thomas, 2013*).

Office workers constitute the largest single occupational sector in developed countries (US Bureau of Labor Statistics); the main risk factor they are all exposed to is the visual display terminal (VDT).

Different biomarkers of alcohol consumption are used in clinical practice to evaluate the patient's alcohol use history. Laboratory markers give objective information about alcohol consumption and changes in consumption over time (some of them are sensitive to a recent assumption and others to a long-term use) (*Sharpe, 2001*).

Gamma-glutamyltransferase (GGT) is a transferase that catalyzes the transfer of gamma-glutamyl functional groups. It is present in the cell membranes of many tissues, but is predominantly used as a diagnostic marker for liver disease (*Tate & Meister, 1985*). An isolated elevation or disproportionate elevation of GGT compared to other liver enzymes can indicate alcohol abuse or alcoholic liver disease (*Kaplan, 1985*). Despite its poor specificity, 50 $\pm$ 72% of elevated GGT levels can be explained by excessive alcohol consumption (*Kristenson et al., 1980*).

Alanine transaminase (ALT) and aspartate transaminase (AST) are transaminase enzyme that catalyze a transamination reaction between an amino acid and an $\alpha$-keto acid. ALT and AST are found in plasma and in various body tissues but they are most common in the liver. Serum ALT and AST levels and their ratio (AST/ALT ratio) are commonly measured

clinically as biomarkers for liver health (*Dufour et al., 2000*) and alcoholic liver disease (*Andresen-Streichert et al., 2018*).

Mean corpuscular volume (MCV) is a laboratory value that measures the average size and volume of a red blood cell. It has utility in helping determine the etiology of anemia; in particular, megaloblastic anemia can be caused by folate deficiency, which is linked to chronic alcoholism (*Maner & Moosavi, 2020*).

Carbohydrate deficient transferrin (CDT) is a kind of transferrin (an iron-binding blood plasma glycoprotein that control the level of free iron) that represents less than the 1.6% of the total transferrin found in plasma. As the plasma half-life of CDT is 10–14 days, a raised percentage of CDT is strongly suggestive of chronic excessive alcohol consumption with sensitivity and specificity both approaching 85% (*Bomford & Sherwood, 2014*). In addition, the production of CDT is directly proportional to alcohol intake (*Golka & Wiese, 2004*).

Biomarkers of alcohol consumption and liver function may respond to even rather low levels of ethanol intake in a gender-dependent manner (*Alatalo et al., 2009*), the overall accuracy of Carbohydrate-deficient transferrin (CDT) and Gamma-glutamyltransferase (GGT), appear to be the highest in the detection of problem drinking (*Anttila et al., 2005*).

It is possible to identify patients with moderate or heavy alcohol consumption: heavy consumption is defined as the ingestion of more than 60 grams of alcohol per day (if protracted for 2 weeks or more, it is considered chronic heavy consumption) (*Peterson, 2004*). Many studies, from the end of the 1980s to today, have analyzed the relationship between unemployment and the use of alcohol (*Kerr, Campbell & Rutherford, 1987*; *Crawford et al., 1987*; *Forcier, 1988*; *Lee et al., 1990*; *Gallant, 1993*; *Lester, 1996*; *Popovici & French, 2013*) but no one, to date, has evaluated this association considering the biomarkers of alcohol consumption.

Considering the recent confirmations of the scientific literature about the usefulness of the laboratory tests mentioned in identifying the use of alcohol (*Niemelä et al., 2019*), the purpose of this cross-sectional study was to compare serum biomarkers of alcohol consumption among office workers and unemployed people.

## MATERIALS & METHODS

### Patient selection

This cross-sectional study examined a sample of people who presented for administrative checks (license request, gun license, etc.), from January 2018 to December 2018, to the forensic toxicology laboratory of a large hospital in the city of Rome. These people were surveyed with one questionnaire, collecting information about demographic and job characteristics. The subjects who declared to work in offices and to use a video display terminal (VDT) at work for more than 20 h per week (as defined by current legislation in Italy) were classified as "office workers" (OW); the exposure to VDT has been investigated, because is considered as risky for the health of workers already for a long time (*Lim, Sauter & Schnorr, 1998*). Unemployed people (UP) have been identified as those who declared to be jobless. Housewives and students were considered in the working group as "Others". Finally, OW and UP were included in the study. All participants provided written informed

consent for data collection. This research conforms to the principles of the Declaration of Helsinki, in accordance with the ''Sapienza'' University of Rome's Ethical Commission regulation and with the Italian law; we have communicated to this commission the starting of our observational study (#02/2018 07/01/2018). We excluded subjects with previous or current liver disease, exposure to hepatotoxic drugs, family history of liver disease, risk factors for viral hepatitis (history of previous transfusions or use of hemoderivates, use of narcotic substances, promiscuous use of syringes, sexual contact with known hepatitis carriers), or exposure to hepatotoxic substances (solvents, paints, pesticides, other).

## Sampling methods and biomarkers of alcohol consumption

Carbohydrate-deficient transferrin (CDT), Gamma-glutamyltransferase (GGT), aspartate aminotransferase (AST), alanine aminotransferase (ALT) levels and mean corpuscular volume (MCV) were measured in all subjects (in the morning, at 8 am). The analysis of liver biomarkers (GGT, ALT and AST) was performed by enzymatic test (IFCC) with ILAB 650® instrument (Instrumentation Laboratory—Werfen Group, Barcelona, Spain); they were considered to be elevated if higher than the respective standard thresholds: 55 U/l, 41 U/l, 37 U/l. The analysis of MCV was performed by impedenzometric test with AcT 8® Instrument (Beckman Coulter. Inc, Brea, California, United States) with the range of normal values between 80 and 100 fL. CDT's analysis was performed with capillary electrophoresis Minicap® (Sebia, Paris, France). The cut-off used for CDT was above 1.6% as recommended by the assay manufacturer. The levels of carbohydrate-deficient transferrin (CDT) are widely used to diagnose alcohol-related disorders in clinical, occupational and forensic contexts (*Bortolotti et al., 2018*; *Helander et al., 2016*) because it is an indicator for long-term alcohol consumption and, after discontinuing drinking, the serum CDT levels usually normalize within approximately 2–4 weeks, but it may take even longer (*Jeppsson, Kristensson & Fimiani, 1993*). A shorter half-life has also been described (*Neumann & Spies, 2003*). It is well known that a variety of medical conditions may elevate GGT levels as well as several medications (*Onigrave et al., 2002*). On the contrary, CDT levels are not influenced by common medications or chronic diseases (*Arndt, 2001*).

## Statistical analysis

Quantitative variables were summarized as mean and standard deviation (SD) or median interquartile range (IQR) according to their distribution. Shapiro–Wilks test was performed to evaluate normal distribution of continuous variables. Qualitative variables were summarized as frequency and percentage. Due to the differences in the number of enrolled subjects between study groups (170 OW vs 270 UP) and in order to remove possible selection bias of our convenience sample, a propensity score matching procedure was performed using a multivariable logistic model with an 8:1 greedy matching algorithm with no replacement (*Parsons Lori, 2001*). All baseline variables included in the matching model are presented in Table 1. The adequacy of covariate balance in the matched sample was assessed via standardized mean differences between two groups, with differences of less than 20% indicating a good balance (*Austin, 2009*). Unmatched subjects were discarded from the analysis. Mann–Whitney *U* test was performed to evaluate differences in the levels

**Table 1 Patients' baseline characteristic before and after the matching procedure.**

| | Unmatched | | Matched | | Standardized mean difference |
|---|---|---|---|---|---|
| | Office workers (n = 170) n (%) | Unemployed people (n = 270) n (%) | Office workers (n = 157) n (%) | Unemployed people (n = 157) n (%) | |
| Age mean (SD) | 38.3 (10.0) | 35.7 (13.8) | 38.3 (10.4) | 38.9 (12.4) | −0.05 |
| Males n (%) | 145 (85.2) | 217 (80.3) | 133 (84.7) | 124 (79.0) | −0.16 |
| Females n (%) | 25 (14.7) | 53 (19.6) | 24 (15.3) | 33 (21.0) | 0.16 |
| Propensity score | 0.403 | 0.390 | 0.402 | 0.398 | 0.07 |

of each biomarker between the two groups. Logistic regression models were performed to evaluate if working status (OW vs UP) predicts abnormal levels for each biomarker. We considered as dependent variables the dichotomized values of each biomarker (pathological vs normal) while working status (office workers vs unemployed) was considered as an independent variable. All logistic models were adjusted for propensity score as covariate. The matching procedure was performed in order to remove possible confounders as age and gender that can influence biomarkers' levels. In particular, this is a cross-sectional study and it can be influenced by the selection bias being a convenience sample. Two-tailed $p$ values less than 0.05 were considered significant. Statistical analysis was performed using IBM$^{\text{TM}}$ SPSS$^{\circledR}$ Statistics for Windows v23.0 (Armonk, NY: IBM Corp.).

## RESULTS

In the study were enrolled 440 subjects (170 office workers and 270 unemployed), and 242 subjects were excluded. After the propensity score matching procedure, 314 participants were selected, thereof 157 office workers (OW) and 157 unemployed people (UP). Figure 1 shows the steps of the selection procedure. Groups were homogeneous for age and gender, as demonstrated by a standardized mean difference lower than 0.20 (Table 1). The statistical analysis showed that there were no differences in biomarkers levels between the two groups. The differences of median values of CDT, GGT, AST, ALT, and MCV were not statistically significant, as reported in Table 2. Absolute frequencies of abnormal values of any biomarker were always lower among office workers. Logistic regression models showed that OW negatively predict abnormal biomarker levels. Office workers had a significant negative association with the levels of GGT (OR 0.48; 95% CI [0.28–0.84]), AST (OR 0.42; 95% CI [0.22–0.78]), ALT (OR 0.39; 95% CI [0.23–0.66]), and MCV (OR 0.37; 95% CI [0.19–0.70]). The results of the logistic regression models were reported in Table 3.

## DISCUSSION

Work is an important determinant of health and, in the field of occupational medicine, there is often a tendency to emphasize more the risks than the benefits of work.

Especially among people of lower socioeconomic status, unemployment is linked to less healthy lifestyles, higher prevalence of obesity, low consumption of fruits and
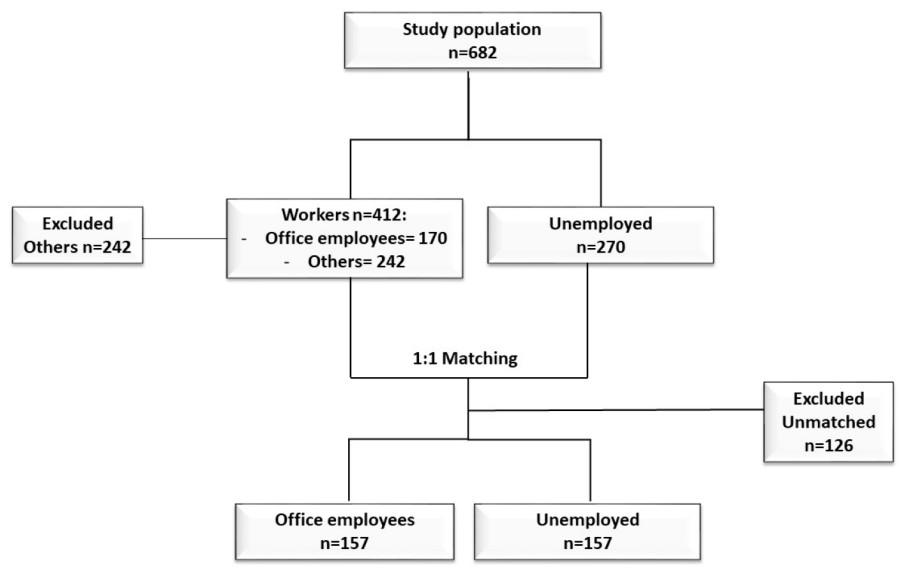

**Figure 1** The steps of the selection procedure.

**Table 2** Differences in biomarker values between office workers and unemployed in matched and unmatched populations.

|  | Unmatched ($n = 582$) | | $p$-value[*] | Matched ($n = 314$) | | $p$-value[*] |
|---|---|---|---|---|---|---|
|  | Office workers ($n = 170$) Median (IQR) | Unemployed ($n = 270$) Median (IQR) |  | Office Workers ($n = 157$) Median (IQR) | Unemployed ($n = 157$) Median (IQR) |  |
| CDT % | 0.7 (0.6–0.8) | 0.7 (0.6–0.9) | 0.736 | 0.7 (0.5–0.8) | 0.7 (0.6–0.8) | 0.350 |
| AST U/L | 19.5 (17.0–22.0) | 19.0 (17.0–23.0) | 0.970 | 20.5 (17.0–25.0) | 20.0 (17.0–23.0) | 0.211 |
| ALT U/L | 20.0 (16.0–27.0) | 19.0 (15.0–26.0) | 0.536 | 19.0 (14.0–30.8) | 20.0 (16.0–26.0) | 0.953 |
| GGT U/L | 20.5 (15.0–29.0) | 20.0 (14.0–31.0) | 0.517 | 21.0 (15.0–32.5) | 20.0 (15.0–29.0) | 0.860 |
| MCV fl | 91.6 (89.8–94.8) | 92.6 (89.7–94.9) | 0.243 | 93.1 (90.0–95.6) | 91.6 (89.9–94.8) | 0.278 |

**Notes.**

IQR, interquartile range; fl, femtolitre; U/L, international Units per litre.

*Mann–Whitney $U$ test.

**Table 3** Logistic regression models evaluating the association between pathological of studied parameters values and office workers.

|  | Office workers ($n = 157$) $n$ (%) | Unemployed ($n = 157$) $n$ (%) | Odd ratio[*] | 95% CI | $p$-value |
|---|---|---|---|---|---|
| CDT | 11 (7.0) | 21 (13.4) | 0.47 | 0.22–1.03 | 0.060 |
| AST | 19 (12.1) | 37 (23.6) | 0.42 | 0.22–0.78 | 0.006 |
| ALT | 30 (34.8) | 127 (55.7) | 0.39 | 0.23–0.66 | 0.001 |
| GGT | 28 (17.8) | 46 (29.3) | 0.48 | 0.28–0.84 | 0.011 |
| MCV | 17 (10.8) | 37 (23.6) | 0.37 | 0.19–0.70 | 0.002 |

**Notes.**

*All models were adjusted for propensity score; Unemployed People were selected as reference.

vegetables and increased consumption of unhealthy foods (*Gallus et al., 2013*; *Dave & Kelly, 2012*). Our results are consistent with the current evidence about the association between unemployment and higher alcohol consumption (*Dom et al., 2016*). Quite recently, this finding has also been confirmed by a large European study (*Bosque-Prous et al., 2015*).

Additionally, unemployment increases the risk for binge drinking, as well as death or hospitalization related to alcohol consumption (*Popovici & French, 2013*; *Czapla et al., 2015*). These data are also consistent with other studies conducted outside the EU (*Midanik & Clark, 1995*; *Cooper, 2011*).

The biomarkers considered in this study are widely used in science and forensics to test for alcohol misuse (*Andresen-Streichert et al., 2018*). In this study, both direct and indirect markers were used. Direct markers are produced when ethanol is metabolized or reacts with the body while indirect markers are enzymes released from dead liver cells following acute or chronic alcohol consumption. The CDT remains in the normal range with a moderate consumption pattern but assumes pathological values with an alcohol intake of more than 50-80 grams of ethanol per day over a period of 1 to 2 weeks (*Helander, 2003*). An increase in MCV, AST, ALT, and GGT values may indicate hepatic damage because of excessive alcohol consumption (*Jastrzębska et al., 2016*) and these indirect parameters take a long time to return to baseline. They are very sensitive but less specific than CDT (*Andresen-Streichert et al., 2018*).

The strength of the study was the propensity score matching procedure that made study groups comparable in terms of baseline characteristics, minimizing possible confounders and the risk of bias in a convenience sample. In addition, the study is based on objective serological data instead of self-reported questionnaires, reducing the risk of bias.

The results of this study must be interpreted considering certain limitations. In fact, we could not match more than one unemployed participant to each office worker, due to the small sample size. In addition, the nature of this study does not allow us to establish a causative relation between alcohol consumption and unemployment or vice-versa. This study also did not consider other possible confounders that might influence the matching procedure.

## CONCLUSIONS

This study showed that office workers had lower absolute frequencies of pathological values of alcohol consumption biomarkers compared to unemployed people, after matching for age and gender. In addition, office work negatively predicts pathological values of alcohol consumption biomarkers.

Office employment seems to be a protective factor against the increase in serum markers of alcohol misuse, compared to unemployed participants.

Surely further studies are needed, but our contribution has shown, on the one hand, that the work itself can represent a protective factor against the use of alcohol and, on the other hand, that this relationship can be highlighted through objective variables such as the blood levels of biomarkers of alcohol-induced liver injury.

### Funding
The authors received no funding for this work.

### Competing Interests
The authors declare there are no competing interests.

### Author Contributions
- Simone De Sio conceived and designed the experiments, performed the experiments, analyzed the data, prepared figures and/or tables, authored or reviewed drafts of the paper, supervision of the manuscript, and approved the final draft.
- Roberta Tittarelli performed the experiments, analyzed the data, prepared figures and/or tables, and approved the final draft.
- Giuseppe Di Martino performed the experiments, prepared figures and/or tables, and approved the final draft.
- Giuseppe Buomprisco and Guglielmo Bruno analyzed the data, prepared figures and/or tables, check references, and approved the final draft.
- Roberto Perri and Fabrizio Cedrone analyzed the data, prepared figures and/or tables, and approved the final draft.
- Flaminia Pantano performed the experiments, analyzed the data, authored or reviewed drafts of the paper, and approved the final draft.
- Giulio Mannocchi performed the experiments, analyzed the data, authored or reviewed drafts of the paper, check references, and approved the final draft.
- Enrico Marinelli performed the experiments, authored or reviewed drafts of the paper, and approved the final draft.

### Human Ethics
The following information was supplied relating to ethical approvals (i.e., approving body and any reference numbers):

Since in Italy the approval of the Ethical Commission is mandatory only for clinical trial, we just communicated to the Ethical Commission of Sapienza the starting of our observational study, as required by the Italian law (#02/2018 07/01/2018).

### Data Availability
The anonymized raw data is available as Supplemental File.

### Supplemental Information
Supplemental information for this article can be found online at http://dx.doi.org/10.7717/peerj.8774#supplemental-information.

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
