# Peer review of "Alcohol consumption and employment: a cross-sectional study of office workers and unemployed people"

_PeerJ, doi:10.7717/peerj.8774_

## Round 0.1 · original submission · Major Revisions

Thank you for your submission. The reviewers have a number of concerns that should be addressed in your revised version.

Reviewer 1 ·

Basic reporting

I was uncertain where the death numbers came from, typically citations come after the information given, I assume perhaps it came from WHO since they were cited at the end of the paragraph on lines 85-89. Perhaps a citation after the first sentence would alleviate this.

Experimental design

First scope of the journal well.

Research question was stated, but I'm not sure how it fills in any gaps unless there have not been other studies to identify actual biomarker differences in alcohol use among the employed and unemployed. The health economics literature has plenty of identification of this difference from self reported alcohol use and alcohol expenditure data, I would highlight the contribution here is the actual biological measure of alcohol use.

The statistical analysis could use more support. The questionnaire was said to have health status and sociodemographic data (on lines 118-119), however the authors only chose to use gender and age in their matching procedure, if you had other covariates why did you not include them? At least give support for why they weren't relevant or exactly what other data you had on these individuals. If age and gender were the only information you had, or you were concerned with standard errors becoming too large by including other controls then give a reasoning why you chose the two you did (mentioned on lines 161-162). I would think gender would be a worse predictor of the treatment than something like health status would be, or other sociodemographic data.

Validity of the findings

The paper did a great job in the conclusion explaining the limitations of the results as causal, and highlighted the real finding of this research which is simply the difference in biomarker measures between those employed and unemployed.

Additional comments

I think the introduction was a good motivator, however, I think it could have had more of the discussion found in the "discussion" section from lines 202-219. It wasn't as clear where this research fit into the bigger pictures in the introduction, but once I read the discussion section it was more clear. This is personal preference of where I would suggest to move that discussion.

I would think you could be worried about differences in the biomarker levels of your excluded sample. Table 1 helps alleviate concerns that the covariates and propensity score of the treatment are similar across your unmatched and matched sample, but if those excluded (which are primarily t hose in the actual treatment of unemployed) had lower biomarker levels then I would be more skeptical of your results. Perhaps including additional rows, or mentioning whether there are significant differences in the Results section would alleviate these concerns.

Similar concerns to my previous comment could be alleviated by providing results or mentioning the how results differ if with replacement is used. This allows you to use the entire sample, and I realize this may increase standard errors to the point you can't find significant differences but if that it is the case I think that is important to note.

·

Basic reporting

In the article "Alcohol consumption and employment: a cross-sectional study of office workers and unemployed people," the authors investigate multiple bio markers associated with alcohol use and employment. They found 4 of 5 markers were associated with significantly decreased odds of being an office worker. The investigation of how employment is associated with substance use is an important contribution to the literature as the protective factors of employment in addiction recovery are not well understood. There is a good deal of research demonstrating substance use disorders are associated with poor employment outcomes. The strength of this paper is the connection with measurable bio markers of alcohol use. However, I have identified several areas in need of improvement or clarification, that if addressed, would improve this manuscript.

1. The authors have presented a considerable amount of information pertaining to the social and physical costs of alcohol misuse and its consequences. However, this information makes up about 1/3 of the introduction. The problems associated with alcohol use are well known and accepted. Yet, the bio markers associated with alcohol-related unemployment are not understood. As this is the real novelty of the study, I suggest the authors provide much more evidence in the introduction that suggest the presently tested bio markers are worthy of investigation in connection with employment. Why were each marker included in the study? What existing research would suggest they are important and associated with alcohol-related unemployment? As it stands, the reader has no idea about the included predictor variables until the methods section, where they seem a bit of a surprise.

2. The research cited about employment and alcohol seems to present how alcohol causes employment. This is a common theme in the literature. However, this can be easily addressed by avoiding causal language such as "affect". Please use words of association, such as related, associated, etc. For example, in the introduction the authors write, “Several studies have found that alcohol abuse negatively affects employment…” Further, there is some new literature that supports employment as a source of "recovery capital" and may support recovery. Please see recovery capital literature, as this may be aligned with the authors' hypotheses. (see Hennessy, 2017, Recovery capital: a systematic review of the literature).

3. Do not use the terms "abuse", "abuser". These terms perpetuate negative stigma and are out of the current trend in addictions literature. Instead, the terms "alcohol use", "alcohol misuse", "alcohol use disorder" (DSM5), and alcohol dependence (ICD-10) are preferred. When referring to a patient, please use person first language; "a person who misuses alcohol". These terms occur throughout the manuscript.

4. I am a bit lost by the presentation office workers use of VDT. It seems that the authors are suggesting the using a computer is associated with greater alcohol consumption. Please give more rationale for the population selection. Additionally, please explain and support with evidence, the connection to the VDT. This is all new information for me so I need more context. I am assuming that other readers may feel the same.

5. It's a bit odd to have the limitations as part of the conclusion. I suggest placing them in the discussion and leave the conclusion for a summary and take-home message.

6. References are listed by name in the text, but by number in the reference list. I suggest using PeerJ style.

7. Table 2. - Decimals are used in the other tables. Please make Table 2 uniform to the others.

8. Please add page numbers

9. Table 3 - Please make clear the number of participants included in the regression analysis. The authors stated, "Unmatched subjects were discarded from the analysis," but for Table 3 Office workers n=105 and Unemployed=268 which does not fit the equal 157 in matched pairs. Additionally, I had to total Ns. Please put Ns in the column headings. Again this may be better explained in the analysis section because I assumed only the matched participants were included in the regression.

Experimental design

1. There is quite a difference in being unemployed and seeking, and unemployed due to being a housewife or student. Finding work is often detrimental for students and housewives. In fact, being a student is associated with greater treatment outcomes (Sahker et al., 2015). I am not convinced the distinction between employed and unemployed is justified. Please either (a) remove students/housewives from the analysis, (b) make a third group, or (c) provide a justification for the inclusion of housewives and students in the unemployed group.

2. Please specify the outcome variable clearly in the methods and explain why it was selected. Currently, the reader must assume you are predicting office worker status from 6 biomedical markers, but this is not specifically stated.

3. Please discuss the full logistic model including all predictors. Is it one model with only the 5 bio markers included? Did you control for other variables such as age, sex, ethnicity, etc.?

4. The statistical analyses seem sound. However, a bit more explanation of a few items would strengthen the paper. I'm not understanding the need to match participants in a regression analysis. The Greedy matching algorithm with may be a bit new to most readers. Please explain how and why participants were matched. Additionally, removing extremes from the sample can add bias. The Greedy method is said to reduce bias and Parsons explains this, but more information is needed in this paper to avoid reader confusion.

5. I'm left wondering if students and housewives were a significant proportion of the unmatched pairs.

Validity of the findings

1. There may be a bit of a misunderstanding here. The authors state, "In this study, office employment seems to be a protective factor against the increase in serum markers of alcohol consumption compared to unemployed participants." However, based on how it is explained, I thought the analysis was a multivariable logistic regression with 5 bio markers predicting 1=employed 0=unemployed. If this is indeed true, I'm not sure it makes sense to state employment protects against alcohol use. Rather, alcohol use bio markers predict employment status. If I am incorrect in my assumption, perhaps more explanation in required. Alternatively, the authors interpretation should be adjusted to represent the data.

2 . Please give some suggestion of how the information found from this study is useful for research and clinical applications.

Additional comments

I found this research to be interesting. I believe that it is an important contribution. I think the majority of issues came from a lack of connection between bio markers, alcohol, and employment. Most of this can be easily addressed with citing evidence and explaining connections. Further, there are some methodological issues that need to be addressed. And finally, I think the paper is missing the "so what?" statement. How does this research fill a gap in the literature and why is is useful. I believe addressing these comments will improve this manuscript.

---

## Round 0.2 · Minor Revisions

Thank you for your revised submission. There are a number of amendments to make before we can consider this for publication. In particular you need to add further detail to your methodology. The reviewer has highlighted a number of areas that require additional clarification.

·

Basic reporting

I had several points of feedback in the first review and the authors gave the same response at least 3 times: “the logistic regression was performed considering the dichotomized values of each biomarkers (pathological vs normal) as a dependent variable and working status (office workers vs unemployed) as an independent variable.”
At times, adequate responses were given in their rebuttal, but they were not entirely incorporated into the manuscript. A good portion of my initial comments still stand. Overall, there just needs to be more and better descriptions of the methods so readers understand the procedures and so the study could be replicated. Additionally, reviewer responses that were addressed, should be done so in the manuscript to improve reader understanding.

1. The authors have presented a considerable amount of information pertaining to the social and physical costs of alcohol misuse and its consequences. However, this information makes up about 1/3 of the introduction. The problems associated with alcohol use are well known and accepted. Yet, the bio markers associated with alcohol-related unemployment are not understood. As this is the real novelty of the study, I suggest the authors provide much more evidence in the introduction that suggest the presently tested bio markers are worthy of investigation in connection with employment. Why were each marker included in the study? What existing research would suggest they are important and associated with alcohol-related unemployment? As it stands, the reader has no idea about the included predictor variables until the methods section, where they seem a bit of a surprise.
ANSWER: We better explained the relationships between alcohol consumption and unemployment and between alcoholism and serum biomarkers of liver dysfunction.
RE-REVIEW: Authors partially responded with, “Biomarkers of alcohol consumption and liver function may respond to even rather low levels of ethanol intake in a gender-dependent manner (Alatalo et al., 2009), the overall accuracy of Carbohydrate-deficient transferrin (CDT) and Gamma-glutamyltransferase (GGT), appear to be the highest in the detection of problem drinking (Anttila et al., 2005).” Readers may assume they randomly chose the other 3 biomarkers since there is nothing in the literature to suggest they are useful. Please properly set up the analysis with background information on all variables used in the model.


2. The research cited about employment and alcohol seems to present how alcohol causes employment. This is a common theme in the literature. However, this can be easily addressed by avoiding causal language such as "affect". Please use words of association, such as related, associated, etc. For example, in the introduction the authors write, “Several studies have found that alcohol abuse negatively affects employment…” Further, there is some new literature that supports employment as a source of "recovery capital" and may support recovery. Please see recovery capital literature, as this may be aligned with the authors' hypotheses. (see Hennessy, 2017, Recovery capital: a systematic review of the literature).
ANSWER: We thank the reviewer for the comment and the precious suggestions. We want to clarify that the aim of the study was the evaluation of the association between working status (office workers vs unemployed) and pathological level of biomarker (as dependent variables). We are in accord with the reviewer about the misuse of the term “affect” and similar because Logistic regression analyses and the cross sectional design do not permit to assume the causal effect. According to these suggestions, we modify the paper.
RE-REVIEW: I am still seeing phrases like “directly affecting the productivity…” please use a term of association here.
This phrase is not clear: “alcohol misuse negatively involves employment.” Do you mean, “alcohol misuse is associated with negative employment outcomes?” and rather than stating, “by directly affecting the productivity and increasing the risk for injuries …” do you mean, “misuse is associated with reduced productivity and increased risk of injuries?”

4. I am a bit lost by the presentation office workers use of VDT. It seems that the authors are suggesting the using a computer is associated with greater alcohol consumption. Please give more rationale for the population selection. Additionally, please explain and support with evidence, the connection to the VDT. This is all new information for me so I need more context. I am assuming that other readers may feel the same.
ANSWER: In order to uniquely characterize the office workers, the exposure to the VDT was chosen to obtain a homogeneous population. Most of the European workforce, in fact, is active in the service sector, as also mentioned in lines 101-103, and the exposure to VDT represents a factor that unites these workers.
RE-REVIEW: Please explain this more clearly in the methods section and not just the response letter. It was confusing to me, so it may be confusing to others.

Experimental design

1. There is quite a difference in being unemployed and seeking, and unemployed due to being a housewife or student. Finding work is often detrimental for students and housewives. In fact, being a student is associated with greater treatment outcomes (Sahker et al., 2015). I am not convinced the distinction between employed and unemployed is justified. Please either (a) remove students/housewives from the analysis, (b) make a third group, or (c) provide a justification for the inclusion of housewives and students in the unemployed group.
ANSWER: We are sorry for the mistake, in line 122 we wanted to state that we considered only jobless as unemployed excluding housewives and students. As reported in Figure 1 (added in the revised manuscript) housewives and students were considered as “other work conditions” and were excluded from the analysis.
RE-REVIEW: In the revised manuscript, the authors wrote, “unemployed people (UP) have been identified as those who declared to be jobless. Housewives and students were considered in the working group as “Others”. Only OW were included in the study.” But, I’m pretty sure both OW and UP were included in the study. Please revise.

3. Please discuss the full logistic model including all predictors. Is it one model with only the 5 bio markers included? Did you control for other variables such as age, sex, ethnicity, etc.?
ANSWER: We thank the reviewer for the comment. Due to the small number of subjects with pathological biomarker’s levels (dependent variables), we were not able to perform a multivariate analysis with all biomarkers. All presented models (one for each biomarker) were only adjusted for propensity score. The propensity score matching was performed to remove the effect of age and gender on study variables.
RE-REVIEW: This still needs to be clearly explained in the manuscript, not just the response. I am not entirely clear on your methods. I think is you conducted 5 regressions to generate a propensity score, then 5 separate logistic models were preformed, each controlling for only the model propensity matching score. Please explain your methods with enough detail.

4. The statistical analyses seem sound. However, a bit more explanation of a few items would strengthen the paper. I'm not understanding the need to match participants in a regression analysis. The Greedy matching algorithm with may be a bit new to most readers. Please explain how and why participants were matched. Additionally, removing extremes from the sample can add bias. The Greedy method is said to reduce bias and Parsons explains this, but more information is needed in this paper to avoid reader confusion.
ANSWER: We thanks the reviewer for the comment. The matching procedure was performed in order to remove possible confounders as age and gender. It’s known that these factors can influence biomarker’s levels. In particular, this is a cross-sectional study and it can be influenced by the selection bias being a convenience sample. In order to improve readability, we modified the methods section.
RE-REVIEW: Please state this response in the manuscript.

Validity of the findings

1. There may be a bit of a misunderstanding here. The authors state, "In this study, office employment seems to be a protective factor against the increase in serum markers of alcohol consumption compared to unemployed participants." However, based on how it is explained, I thought the analysis was a multivariable logistic regression with 5 bio markers predicting 1=employed 0=unemployed. If this is indeed true, I'm not sure it makes sense to state employment protects against alcohol use. Rather, alcohol use bio markers predict employment status. If I am incorrect in my assumption, perhaps more explanation in required. Alternatively, the authors interpretation should be adjusted to represent the data.
ANSWER: As previously stated, the logistic regression was performed considering the dichotomized values of each biomarkers (pathological vs normal) as a dependent variable and working status (office workers vs unemployed) as an independent variable. The aim of the study was to assess the association between working status and pathological biomarker levels and not vice-versa, so in the light of this clarification, our statement is correct. However, we clarify this point in the paper.
RE-REVIEW: Perhaps this is just stylistic, but when the authors use the term “association” when referencing a regression analysis, it suggests a mutual relationship. They are not wrong for using this term, but it is not entirely accurate. However, if they were to say “working status predicts biomarker levels,” it would be clear that the biomarkers are the dependent variables.

---

## Round 0.3 · accepted · Accept

Thank you for your updated submission. I am pleased to tell you that this has now been accepted for publication.

·

Basic reporting

no further comment

Experimental design

no further comment

Validity of the findings

no further comment

Additional comments

The authors have done a good deal of work addressing feedback, which I believe has improved the paper. I am very satisfied with their edits and responses to the feedback.